# Effect of Silver Modification on the Photoactivity of Titania Coatings with Different Pore Structures

**DOI:** 10.3390/nano11092240

**Published:** 2021-08-30

**Authors:** Borbála Tegze, Emőke Albert, Boglárka Dikó, Norbert Nagy, Adél Rácz, György Sáfrán, Attila Sulyok, Zoltán Hórvölgyi

**Affiliations:** 1Centre for Colloid Chemistry, Department of Physical Chemistry and Materials Science, Faculty of Chemical Technology and Biotechnology, Budapest University of Technology and Economics, H-1521 Budapest, Hungary; tegze.borbala@vbk.bme.hu (B.T.); albert.emoke@vbk.bme.hu (E.A.); dbogi95@gmail.com (B.D.); 2Centre for Energy Research, Institute of Technical Physics and Materials Science, H-1121 Budapest, Hungary; nagy.norbert@energia.mta.hu (N.N.); racz.adel@energia.mta.hu (A.R.); safran.gyorgy@energia.mta.hu (G.S.); sulyok.attila@energia.mta.hu (A.S.)

**Keywords:** titania, sol-gel coating, photocatalysis, self-cleaning, dye degradation, dye association

## Abstract

Nanostructured photoactive systems are promising for applications such as air and water purification, including self-cleaning coatings. In this study, mesoporous TiO_2_ sol-gel coatings with different pore structures were prepared and modified with silver by two methods: the “mixing” method by adding AgNO_3_ to the precursor sol, and the “impregnation” method by immersing the samples in AgNO_3_ solution (0.03 and 1 M) followed by heat treatment. Our aim was to investigate the effects that silver modification has on the functional properties (e.g., those that are important for self-cleaning coatings). Transmittance, band gap energy, refractive index, porosity and thickness values were determined from UV-Vis spectroscopy measurements. Silver content and structure of the silver modified samples were characterized by X-ray photoelectron spectroscopy, Rutherford backscattering spectrometry, High-resolution transmission electron microscopy and Energy Dispersive X-ray Spectrometry elemental mapping measurements. Wettability properties, including photoinduced wettability conversion behavior were investigated by water contact angle measurements. Photoactivity was studied under both UV and visible light with rhodamine 6G and methylene blue dye molecules, at the liquid–solid and air–solid interfaces modeling the operating conditions of self-cleaning coatings. Samples made with “impregnation” method showed better functional properties, in spite of their significantly lower silver content. The pore structure influenced the Ag content achieved by the “impregnation” method, and consequently affected their photoactivity.

## 1. Introduction

Nanostructured semiconductor (e.g., TiO_2_) based systems that show photocatalytic activity are promising for applications such as solar cells, gas sensing, and air and water purification, including self-cleaning coatings [1,2,3]. TiO_2_ is an efficient photocatalyst, and its photoactivity can be further increased by modification with noble metals, e.g., silver. Silver particles can inhibit the electron-hole recombination process by formation of a Schottky barrier at the metal–semiconductor interface, thus leading to faster degradation of organic molecules at the titania surface [1,4]. TiO_2_–Ag composite materials show high photoactivity and antibacterial property, and they are promising for many applications, such as photocatalysts for waste water treatment and other catalytic applications, self-cleaning coatings and photoanodes in dye sensitizes solar cells (DSSCs) [5,6,7,8,9,10,11,12].

Photocatalysts can be characterized by dye degradation under UV (Ultra Violet) or visible light. In most experiments, the photodegradation process is investigated at the liquid–solid interface, in dye solutions [13,14,15,16]. However, in order to model the operating conditions of self-cleaning coatings, experiments at the air–solid interface are also very important, and less research has been conducted on this topic [17,18,19]. Studying the photoinduced wettability change of titania-based photocatalysts is also important: this commonly observed property of TiO_2_ means that complete wetting (contact angle equals zero) can happen due to UV irradiation, and thus rain will facilitate the self-cleaning process in outdoor applications [10,20].

Mesoporous TiO_2_ coatings can be prepared by the sol-gel method. The simplest and most often used way to prepare TiO_2_–Ag composites with this method is to add AgNO_3_ (or other precursor containing silver ions) to the precursor sol [21,22,23]. Another promising possibility is to modify the as-prepared mesoporous coatings by impregnation in AgNO_3_ solution, followed by post-treatments step(s) in which the silver ions are reduced [24,25,26,27]. These post-treatment steps most often include heat treatment [24,25] or UV irradiation [26,27]. Some further examples of less often used methods for silver modification include using a microemulsion system [28], and electrochemical deposition [29].

We report in this paper, mesoporous TiO_2_–Ag sol-gel coatings with different pore structures were prepared using pore forming templates. The Ag modification was carried out using both methods mentioned above. Our aim is to seek a better understanding of the effects that silver modification has on the functional properties of titania, especially those properties that are important for self-cleaning coatings. The advantages and disadvantages of the different silver modification methods, and also the effect of the silver content on the photoactivity are investigated. The photoactivity was studied under both UV and visible light with different dye molecules. The dye degradation was studied at the air–solid and liquid–solid interfaces modeling the operating conditions of self-cleaning coatings. Furthermore, the model systems investigated in our studies have relevance in the fields of DSSCs, and also the treatment of waste waters from the textile industry, that contain dye molecules. Dye association was studied in detail, as per our previous findings [18] such processes can have an important role in the photodegradation test results. The photoinduced wettability conversion behavior of the samples was also studied.

## 2. Materials and Methods

The following materials were used in the experiments: ethanol (EtOH, 99.7+%, Reanal, Budapest, Hungary), 2-propanol (99+%, Acros Organics, Geel, Belgium), purified distilled water (filtered with Millipore Simplicity 185 system to reach 18.2 MΩ cm), tetraethyl orthosilicate (TEOS, 99%, Merck, Budapest, Hungary), titanium (IV) butoxide (98%, Merck, Budapest, Hungary), titanium (IV) isopropoxide (98+%, Acros Organics, Geel, Belgium), hydrochloric acid (HCl, 37%, Reanal, Budapest, Hungary), nitric acid (HNO_3_, 65%, Lach-Ner, Bratislava, Slovakia), ammonia aqueous solution (NH_3_, 25%, Lach-Ner, Bratislava, Slovakia), acetylacetone (99+%, Acros Organics, Geel, Belgium), hexadecyltrimethylammonium bromide (CTAB, 99+%, Acros Organics, Geel, Belgium), Pluronic P123 (P123, average mol wt 40000, Sigma Aldrich, Budapest, Hungary), AgNO_3_ (99.8%, Lach-Ner, Bratislava, Slovakia), rhodamine 6G (R6G, 95%, Sigma Aldrich, Budapest, Hungary), methylene blue (MB, 95+%, Reanal, Budapest, Hungary) and methyl orange (MO, 95%, Acros Organics, Geel, Belgium) dyes. All materials were analytical grade and used without further purification.

### 2.1. Preparation of Coatings

TiO_2_ coatings of different pore structures were prepared on glass, quartz glass and Si substrates by sol-gel spin-coating method. The glass substrates were first coated with a compact SiO_2_ barrier layer, in order to inhibit Na^+^ diffusion from the glass into the titania layer, which would decrease the photoactivity [30]. TiO_2_–Ag composite coatings were prepared by two different silver modification methods: in the “impregnation method” the TiO_2_ coatings were impregnated in AgNO_3_ solution, then a heat treatment was carried out to reduce the silver ions; in the “mixing method” the AgNO_3_ was added to the precursor sol before the film deposition.

Precursor sols were synthesized by the acid catalyzed hydrolysis and polycondensation of metal-alkoxides as reported in our previous publications [18,31]. SiO_2_ barrier layers were prepared from precursor sols with molar ratios of TEOS: EtOH:HCl:H_2_O—1:4.75:7.2 × 10^−4^:4.00. Two types of porous TiO_2_ coatings were prepared using two different pore forming agents. Samples prepared with CTAB as template denoted c-TiO_2_ were made from precursor sols with the molar ratios of titanium (IV) butoxide:EtOH:HNO_3_:H_2_O:CTAB were 1:27.74:0.39:1.54:0.124 in the precursor sol. Samples prepared with P123 as template (denoted p-TiO_2_) were made from precursor sols with the molar ratios of titanium (IV) isopropoxide:EtOH:acetylacetone:H_2_O:P123 were 1:33.80:1.93:4.39:0.034, while silver modified composite coatings denoted p-TiO_2_–Ag were prepared from similar precursor sols that additionally also contained AgNO_3_, with a molar ratio of 0.10.

Film deposition from the precursor sols onto clean glass and Si substrates (cleaned with 2-PrOH and distilled water) and SiO_2_ coated glass substrates was carried out with a spin-coater (SCS 6808 Spin Coater) using spinning speeds of 8500 rpm for SiO_2_ barrier layers, and 6000 rpm for TiO_2_ coatings. Samples were heated in an oven (Nabertherm B170) for 1 h at 480 °C (p-TiO_2_ and p-TiO_2_–Ag coatings), or for 30 min at 450 °C (SiO_2_ and c-TiO_2_ coatings). Silver modified TiO_2_ coatings were also prepared by impregnation of TiO_2_ samples in AgNO_3_ aqueous solutions of 0.03 M or 1 M concentrations. The impregnation was carried out with a dip-coater (Plósz Mérnökiroda Kft., Hungary) using a dipping speed of 1 cm min^−1^ and withdrawal speed of 12 cm min^−1^. The samples were then rinsed with distilled water, dried, and thermally treated at 60 °C for 30 min, followed by 450 °C for 30 min. The coatings prepared with this impregnation method were denoted as c(or p)-TiO_2_(0.03Ag) and c(or p)-TiO_2_(1Ag) depending on the applied 0.03 and 1 M concentrations, respectively.

### 2.2. Characterization

The transmittance spectra of the glass substrates and the coated samples were measured by an Analytic Jena Specord 200–0318 UV-Vis spectrophotometer, in the wavelength range of 350–1100 nm, using air as reference, a scanning speed of 10 nm s^−1^ and a resolution of 1 nm. The refractive index (at 632.8 nm) and layer thickness values of SiO_2_, and SiO_2_/TiO_2_ two-layered coatings on glass substrates were calculated from the transmittance spectra using thin layer optical models [32]. The porosity of these coatings was estimated from the measured effective refractive index using the Lorentz–Lorenz formula [33].

High-resolution transmission electron microscopy (HRTEM) measurements and energy dispersive X-ray spectrometry (EDS) elemental mapping of p-TiO_2_, p-TiO_2_(0.03Ag), p-TiO_2_(1Ag) and p-TiO_2_–Ag coatings on Si substrates were carried out, in cross section view, by means of a 200 kV C_s_-corrected FEI Themis scanning-transmission electron microscope.

The surface composition of the coatings was investigated by X-ray photoelectron spectroscopy (XPS), using a special cylindrical mirror analyzer (type DESA 105, Staib Instruments Ltd., with detector resolution of 1.5 eV) with retarding field and Al anode X-ray source. For cleaning the surface from air contamination and revealing the subsurface composition as well, the samples received modest argon sputtering (1 keV, 75° respect to the surface normal), and spectra were taken before and after sputtering as well. The background pressure was 2 × 10^−9^ mbar which was increased up to 2.5 × 10^−7^ mbar during argon ion bombardment. The following main XPS lines have been recorded: O-1s at 531.7 eV; Na-KLL Auger at 497.4 eV; Ti-2p 3/2 at 454 eV; Ag-3d 5/2 at 368.4 eV; C-1s at 284.8 eV; Si-2p at 99.8 eV (with 0.1 eV steps and 1000 ms dwell time per data point). Spectra were evaluated by CasaXPS program. The energy shift of peaks because of charging was corrected by the adventitious carbon C-1s peak using 284.8 eV reference value. A Gaussian–Lorentzian peak shape was used for peak decomposition. The elemental compositions have been determined from peak areas assuming homogeneous model after Shirley background subtraction, and the sensitivity factors were taken from Reference [34]. The determined at% values show the average chemical surface composition of a 5 mm diameter area with an information depth of 3 nm.

Rutherford backscattering spectrometry (RBS) measurements of different Ag modified p-TiO_2_ coatings (p-TiO_2_(0.03Ag), p-TiO_2_(1Ag), p-TiO_2_–Ag) on SiO_2_ coated glass, and of reference samples (glass, SiO_2_ coated glass, p-TiO_2_ on SiO_2_ coated glass) were applied to determine the silver content of the samples. Measurements were carried out in a scattering chamber with a two-axis goniometer connected to a 5 MV Van de Graaff accelerator. The 1 MeV ^4^He^+^ analyzing ion beam was collimated with two sets of four-sector slits to the dimensions of 0.5 mm × 0.5 mm, while the beam divergence was kept below 0.06°. The beam current was measured by a transmission Faraday cup. In the scattering chamber the vacuum was about 1 × 10^−4^ Pa. To reduce the hydrocarbon deposition, liquid N_2_ cooled traps were used along the beam path and around the wall of the chamber. Backscattered He^+^ ions were detected using an ORTEC surface barrier detector mounted in Cornell geometry at scattering angle of 165°. The energy resolution of the detection system was 16 keV. Spectra were recorded at sample tilt angles of 7° and 60°. In all experiments both axial and planar channeling of the He^+^ projectiles in the single crystalline Si substrate was avoided. The measured RBS spectra were evaluated with the RBX spectrum simulation code.

Advancing and receding water contact angles were measured on the coatings by a drop shape analyzer device (DSA30, KRÜSS GmbH, Hamburg, Germany). The measurements were made with distilled water, using the sessile drop method and the drop-build-up technique. The coatings were placed in a closed chamber (relative humidity: above 90%, temperature: 25 °C), then a water droplet of 10 µL was placed on the surface to measure the advancing contact angles (Θ_A_), then 5 µL of water was removed to measure the receding angle (Θ_R_), and this was repeated two more times for each sample. Contact angle hysteresis (H) was determined as H = Θ_A_ − Θ_R_, and Young’s contact angle (Θ_Young_) was calculated using the Wolfram-Faust equation [35]: Θ_Young_ = arccos(cos Θ_A_ + cos Θ_R_)/2). The photoinduced surface wettability conversion of the samples was studied by placing the samples under visible (two GE 93,845 57 W LED lamps, emission maximum at 600 nm, distance between light source and samples: 18 cm, light intensity: 0.47 mW cm^−2^) and UV light sources (Phillips CLEO HPA 400 S, 400 W UV-A lamp, emission maximum at 365 nm, distance: 30 cm, intensity: 47.1 mW cm^−2^).

Dye adsorption on the titania surface was investigated in dye impregnation measurements. Samples were impregnated in MB and R6G aqueous dye solutions (10^−3^ M,), using the dip-coater, with dipping and withdrawal speed of 10 cm min^−1^, and impregnation time of 2 min. The pH of the solutions was adjusted with aqueous NH_3_ solution to pH = 10, in order to have stronger electrostatic interactions between the cationic molecules and the surface (as the surface is more negatively charged at higher pH values) [18,19]. After impregnation the samples were rinsed with distilled water, dried at room temperature in air, and their absorbance spectra were measured. As a pre-treatment, before the start of all dye photodegradation tests, silver modified TiO_2_ coatings were irradiated with UV light to reduce any silver ions present on the surface. This UV irradiation resulted in increased absorbance, and 10 min was chosen for the time of the UV treatment, since longer irradiation times caused no significant change in the absorbance spectrum.

Dye photodegradation was studied at air–solid interface with MB and R6G dyes, under both UV and visible light. Two UV-A lamps (Sylvania BL368 20 W, emission maximum at 368 nm) were used as UV light source, with a distance of 18 cm and intensity of 0.20 mW cm^−2^. Two LED lamps (GE 93,845 57 W, emission maximum at 600 nm) were used as visible light source, with a distance of 18 cm and intensity of 0.47 mW cm^−2^. Ag modified TiO_2_ coatings were kept under UV irradiation for 10 min before the start of the photocatalysis measurements. Dye impregnated samples were placed under UV or visible light source with the coated side upwards, and irradiated for different time intervals. After each time interval the absorbance spectra of the coatings were measured. All photocatalysis test results were averaged from data measured on at least three identical samples in each case.

For comparison, dye photodegradation at the liquid–solid interface was also studied on c-TiO_2_, p-TiO_2_, c-TiO_2_(0.03Ag) and p-TiO_2_(0.03Ag) samples. Dye solutions of 10^−5^ M concentration were made with distilled water, and 25–25 mL solutions were placed in crystallizing dishes, and placed under a UV lamp. A Phillips CLEO HPA 400 S, 400 W UV-A lamp (emission maximum at 365 nm, distance: 30 cm, intensity: 47.1 mW cm^−2^) was used as UV light source. One sample was placed in each crystallizing dish, with the coated side upwards. During the measurement the solutions were continuously mixed, while cooled in a water bath and ventilation was also used to counteract the heat generated by the UV lamp. Before the irradiation, the samples were kept in the solutions for 10 min, in order to reach adsorption equilibrium. Absorbance spectra of the solutions were taken before the start of the measurement, after the adsorption step, and after each hour of irradiation, until a total of 4 h UV irradiation was carried out. Absorbance was measured in a quartz cuvette, using distilled water as reference.

## 3. Results

### 3.1. Characterization of TiO_2_ and TiO_2_–Ag Coatings

The TiO_2_ and TiO_2_–Ag composite coatings were characterized by UV-Vis spectroscopy, HRTEM and EDS elemental mapping, RBS, XPS and surface wettability measurements.

Transmittance spectra of the different coatings can be seen in Figure 1a. The refractive index and film thickness values were determined from the transmittance data, and the porosity was estimated from the refractive index values (Table 1). The SiO_2_ barrier layer has a refractive index close to the value found in literature for pure SiO_2_ (1.46 [36]), proving that this coating has a dense, compact structure without pores. Both types of TiO_2_ coatings have a porous structure, and the p-TiO_2_ coatings have higher film thickness and porosity. TiO_2_–Ag composite coatings showed higher light absorbance due to the presence of silver in the coatings.

c-TiO_2_, p-TiO_2_, p-TiO_2_(1Ag) and p-TiO_2_–Ag coatings were prepared on quartz glass slides and their absorbance spectra was measured in order to determine the band gap energy of the titania coatings. The Tauc plots of the samples can be seen in Figure 1b. Indirect band gap was assumed, and the determined band gap energy was found to be similar for c-TiO_2_ and p-TiO_2_ samples (3.39 and 3.38 eV, respectively), while the Ag modified coatings had a slightly lower band gap energy (3.34 eV for p-TiO_2_(1Ag) and 3.33 eV p-TiO_2_–Ag coatings). This suggests that TiO_2_ is present mainly in its anatase crystal phase in these samples, in accordance with the results of our previous publication [25], in which XRD measurements were carried out on model powder samples of the c-TiO_2_, c-TiO_2_(1M), c-TiO_2_(0.03M) and p-TiO_2_–Ag type coatings, and where it was found that anatase phase is present in 90–100 wt %, while rutile phase is present in 0–10 wt %, and the crystallite size was determined to be ~12 nm.

Typical cross-sectional HRTEM images and EDS elemental maps of Ag of p-TiO_2_ type coatings and their silver modified versions can be seen in Figure 2. Ag particles appear, in the EDS elemental mapping images, in red color. Layer thickness values were determined to be 93 ± 4 nm for p-TiO_2_, 96 ± 4 nm for p-TiO_2_(0.03Ag), 99 ± 2 nm for p-TiO_2_(1Ag) and 92 ± 2 nm for the p-TiO_2_–Ag coating. These results are in good agreement with the value determined by UV-Vis spectroscopy for the p-TiO_2_ coatings, and show that Ag modification did not cause significant change in coating thickness. It was found that p-TiO_2_, p-TiO_2_(0.03Ag) and p-TiO_2_(1Ag) samples show a similar porous structure with mostly 5–8 nm sized pores, while the p-TiO_2_–Ag coating has a different, rougher structure made up of bigger particles with pore sizes of 10–15 nm.

Silver particles were detected in all three silver modified samples, but the size and distribution of the particles were found to be different. The p-TiO_2_–Ag coatings contain very small Ag particles (typically 0.5–1 nm in size), which are mostly evenly distributed on the surface, with the exception of a few places where the particles are more densely packed together in 5–13 nm sized domains. The p-TiO_2_(0.03Ag) coating contains relatively big (3–6 nm) Ag particles irregularly distributed in the pore system, while the p-TiO_2_(1Ag) samples contain much smaller (~1 nm) silver particles in a higher amount.

Expected silver content values and those determined from RBS and XPS measurements can be seen in Table 2. When calculating the expected values of the TiO_2_–Ag coatings prepared with impregnation method, it was assumed that the AgNO_3_ solution completely fills up the pore volume; while the silver content of p-TiO_2_–Ag samples was calculated from the AgNO_3_ content of the precursor sol. RBS measurements were made on p-TiO_2_(0.03Ag), p-TiO_2_(1Ag) and p-TiO_2_–Ag samples, while the values for c-TiO_2_(0.03Ag) and c-TiO_2_(1Ag) samples are taken from our previous publication [31]. The composition of all coating types was measured at the surface and in the bulk of the samples by XPS, and the detailed results can be seen in Appendix A. Silver content measured by XPS can be compared with the RBS results and expected values by calculating wt %, as can be seen in Table 2. The silver content wt % values calculated from XPS measurements mostly show good agreement with the RBS results, with the exception of c-TiO_2_(1Ag) samples, which have a higher silver content according to the XPS data. This difference might be because the RBS measurements were made on dip-coated samples in the case of the c-TiO_2_(0.03Ag) and c-TiO_2_(1Ag) coatings, which most likely have a slightly different pore structure than the spin-coated samples. RBS and XPS silver content values are in most cases much higher than the expected values, which suggests that silver ion adsorption on the surface had a significant role during the impregnation. It was found, that p-TiO_2_–Ag coatings have a much higher silver content compared to AgNO_3_ impregnated samples. By analyzing the XPS spectra it was found, that in all TiO_2_–Ag composite coatings, the silver is present both in oxide and metallic form, and the oxide form is more dominant. Furthermore, slightly higher amount of silver is present in oxide form at the surface than in the bulk phase. It was found that p-TiO_2_–Ag samples contain more metallic Ag compared to the impregnated TiO_2_–Ag samples. XPS measurements also showed (Appendix A) that Na^+^ could be detected in all samples, meaning that the SiO_2_ coating was not completely successful in insulating the TiO_2_ layer from the glass. Less Na^+^ was detected in the bulk, showing that the sodium ions were segregated to the surface.

The surface wettability of the coatings was investigated by measuring advancing and receding contact angles of distilled water. Contact angles were measured before and after irradiation with UV or visible light, in order to investigate the photoinduced changes in the wetting properties. The measured values, and the determined hysteresis and Young’s contact angle values can be seen in Appendix A. All samples were hydrophilic, and the measured contact angles changed depending on environmental changes: as-prepared samples showed contact angles below 30°, and this value increased the longer the samples were kept in dark. As a result of UV irradiation the surface became more hydrophilic and contact angles decreased to zero or nearly zero for all samples. After keeping the samples in darkness or under visible light, higher contact angle values of 50–70° could be reached. Most samples behaved similarly with only slight differences, with the exception of p-TiO_2_–Ag samples, which showed generally higher contact angle values, and became less hydrophilic under UV light compared to the other samples. In Figure 3 it can be seen that p-TiO_2_–Ag samples needed much longer UV irradiation time to reach low contact angle values compared to p-TiO_2_ coatings, and even 35 min was not enough to reach complete wetting on the whole surface. Regeneration under visible light also happened slightly slower.

### 3.2. Adsorption and Photodegradation of Dyes in the Coatings

#### 3.2.1. Dye Adsorption

During the dye impregnation measurements the coatings were soaked in aqueous dye solutions of Rhodamine 6G (R6G), methylene blue (MB) and methyl orange (MO) dyes, and in preliminary experiments different pH values were tried. Cationic R6G and MB dyes showed significant adsorption at pH = 10, due to electrostatic attraction to the negatively charged surface. The anionic MO dye could not be adsorbed on the coating surface, not even at lower pH values. Thus, the dye impregnation and the photodegradation at air–solid interface studies were carried out with R6G and MB dyes, while all dyes were measured in the liquid–solid interface photodegradation tests.

The normalized absorbance spectra of R6G dye impregnated samples (seen in Figure 4a,b) show a peak around 536 nm, which can be attributed to the monomer molecules, while the smaller peak around 505 nm can be attributed to the H-dimers [37]. The c-TiO_2_ type coatings contain more dimer molecules compared to the p-TiO_2_ type coatings, due to the differences in pore sizes. Generally, the addition of silver into the pore structure led to a lower amount of dimers present in the dye impregnated coatings. The normalized absorbance spectra of methylene blue dye impregnated coatings (Figure 4c,d) show a monomer peak around 660 nm, while associated forms of the dye (dimers and higher associates) have absorbance in the lower wavelength region between 560–610 nm: the highest peak around 573 nm can be attributed to the trimer form [38]. The p-TiO_2_–Ag coatings contain significantly less dye molecules in trimer form, and these are the only coatings where the MB dimers show an easily observed peak (at 605 nm [38]), that can also be seen on the spectra of the impregnating solution.

Figure 4e shows the maximum absorbance (A(max)) values measured on the dye impregnated samples, which gives information about the dye adsorption capability of the different coatings. The amount of adsorbed dye was found to be similar for most samples, with the exception of p-TiO_2_–Ag samples, which adsorbed less dye molecules compared to the other coatings. In some cases, AgNO_3_ impregnated coatings also showed slightly lower dye adsorption.

#### 3.2.2. Rhodamine 6G Dye Degradation at the Air–Solid Interface

Photoactivity of the samples was studied by photodegradation of different dyes at air–solid interface both under UV and visible light. For comparison, dye photodegradation was also investigated at the liquid–solid interface on some samples.

R6G dye photodegradation at the air–solid interface under UV light was studied by measuring the absorbance spectra of R6G impregnated coatings after certain times of irradiation. Dye decolorization was characterized by A/A_0_ − t curves (Figure 5), where A_0_ and A is the absorbance (at the maximum around 536 nm for monomer degradation, and at 505 nm for dimer degradation) initially and after t time, respectively. The kinetics of dye degradation was also investigated, by applying models from the literature. It was found, that monomer degradation can be described as two consecutive first order reactions (see Equation (1)), and a model first presented by Julson and Ollis could be applied, as described by equation 2 [17,18,19].
(1)$$dye\overset{k_{1}}{\to }intermediate\overset{k_{2}}{\to }colorless\;products$$
(2)$$\frac{A}{A_{0}}=\left(\frac{ak_{m1}}{k_{m2}-k_{m1}}\right)(\exp\left(-k_{m1}t\right)-\exp\left(-k_{m2}t\right))+\exp\left(-k_{m1}t\right)$$
where *a* is the ratio of α molar absorptivities: *a* = α(intermediate)/α(R6G), and *k_m1_*, *k_m2_* are the degradation rate constants (for the first and second interval of degradation, respectively) of the R6G monomer molecule.

However, the Julson–Ollis model equation showed poor fitting to the dimer degradation. In some cases, dimer degradation followed pseudo-first order reaction kinetics, while in other cases this was not true in the initial part of the degradation. For this reason, first order degradation rate constant (*k*) values were determined for the whole dimer degradation (*k_dt_*), and also separately for the first (*k_d1_*, up to 4 min) and second (*k_d2_*, from 6 min) time intervals. For comparison, the monomer degradation rate constants (*k_m1_* and *k_m2_*) were also determined by the first-order model. The degradation rate constant values can be seen in Appendix A.

Similarly to our previous findings [18], the monomer degradation was found to be significantly faster than the dimer degradation, especially in the initial part of the degradation process. The p-TiO_2_ coatings showed higher photoactivity than c-TiO_2_ coatings. Both c-TiO_2_(0.03Ag) and c-TiO_2_(1Ag) samples show increased photodegradation compared to c-TiO_2_ samples. In contrast, p-TiO_2_(0.03Ag) and p-TiO_2_(1Ag) samples show lower photodegradation rates than p-TiO_2_ samples. However, p-TiO_2_–Ag samples showed the highest photodegradation rates out of all samples, especially in the first interval of the degradation.

The characterization of R6G photodegradation under visible light was similar to that described above. *A/A_0_ − t* curves can be seen in Figure 6, and the degradation rate constant values in Appendix A. Higher photodegradation was observed in the case of p-TiO_2_ samples compared to c-TiO_2_ samples, similarly to the results under UV light. Additionally, similar to the results described above, c-TiO_2_ coatings show higher photoactivity with increasing Ag content for the impregnated samples, while the AgNO_3_ impregnation modified p-TiO_2_ samples showed no improvement in the degradation rates (with p-TiO_2_(1Ag) samples even performing significantly worse). In contrast to the previous results, the p-TiO_2_–Ag samples showed the lowest photoactivity under visible light among all silver modified samples.

Quantum efficiency and quantum yield values for the first interval of R6G monomer photodegradation were determined for both UV and visible light tests: the detailed calculation is described in the Appendix A (in Appendix A) and in our previous publication [18], and the results can be seen in Appendix A. Generally, QE and QY values for the UV photodegradation were much higher compared to that of degradation under visible light, due to the different mechanism of dye degradation in these experiments. Under UV, the highest QE and QY values were determined for c-TiO_2_(0.03Ag) and c-TiO_2_(1Ag) coatings, while under visible light the p-TiO_2_(0.03Ag) coatings also showed comparably good results. A marked difference could be seen in the QE and QY values of p-TiO_2_–Ag samples under visible and UV light: the values were found to be much higher in the UV experiments, and very low under visible light.

#### 3.2.3. Methylene Blue Dye Degradation at the Air–Solid Interface

MB shows a much slower degradation under UV light than R6G. Furthermore, no dye degradation was observed under visible light, which means that the MB dye was not capable of sensitizing the TiO_2_ photocatalysts. *A/A_0_ − t* curves of MB photodegradation under UV light can be seen in Figure 7. The photodegradation starts with the quick decrease of the small monomer peak, while the trimer peak increases, showing that the monomer photostability is much lower than that of the associated form, and also that trimerization occurs in the samples during the UV irradiation.

Degradation rate constants of MB monomer were determined by the Julson–Ollis model (see Appendix A). It was found that p-TiO_2_, p-TiO_2_(0.03Ag) and p-TiO_2_(1Ag) samples showed slightly faster monomer degradation rates compared to the other coatings, while p-TiO_2_–Ag showed slightly slower monomer degradation. Trimer degradation kinetics was more complex, mostly due to the initial trimerization process, and neither the Julson–Ollis, nor the pseudo-first order model could be applied. The c-TiO_2_ coatings showed lower monomer photodegradation, but higher MB trimer photodegradation compared to p-TiO_2_ coatings. Ag modification of the samples caused a decrease or no significant change in the monomer degradation rates. The samples with higher Ag content showed lower trimer photodegradation rates, in part due to the initial trimerization under UV light, which process was found to be more prominent in the presence of silver. The p-TiO_2_–Ag coatings showed much lower monomer and trimer photodegradation rates compared to all other samples.

#### 3.2.4. Dye Degradation at the Liquid–Solid Interface, under UV Light

Dye photodegradation under UV light at the liquid–solid interface was measured on c-TiO_2_, c-TiO_2_(0.03Ag), p-TiO_2_ and p-TiO_2_(0.03Ag) samples, using 10^−5^ M aqueous solution of R6G, MB and MO dyes (Figure 8a–c). Kinetics could be described by pseudo-first order model, and the *k* values were determined (Figure 8d). Significantly faster dye degradation was observed in the presence of p-TiO_2_ coatings compared to c-TiO_2_ coatings. The c-TiO_2_ samples showed higher photoactivity after AgNO_3_ impregnation for all the investigated dyes. The difference was smaller in the case of p-TiO_2_ samples, where the Ag modified coatings showed higher photoactivity for MO degradation, but only a slight increase could be observed for R6G and MB degradation. These results are in good agreement with the detailed study of photodegradation at the air–solid interface, described above.

## 4. Discussion

The p-TiO_2_ coatings showed higher photoactivity than the c-TiO_2_ coatings under both UV and visible light. The main reason for this is their different pore structure: p-TiO_2_ contains slightly bigger mesopores, as shown in our previous paper [18]. Furthermore, according to the XPS results (Appendix A), there are significantly more Na ions present in the c-TiO_2_ coatings, which can also contribute to their lower photoactivity. There were differences in the dye adsorption capability and the amount of associated dye molecules present in these two types of coatings (see Section 3.2.1), also caused by the different pore structures, which can also contribute to the differences in the photodegradation test results.

Silver addition to the TiO_2_ coatings had different effects on the photoactivity, depending on the type of coating, the silver modification method, and also the dye and the wavelength of light used in the photodegradation tests.

The degradation rate constants and the quantum efficiency and quantum yield values of the Rhodamine 6G dye photodegradation tests show that under both UV and visible light the c-TiO_2_(0.03Ag) and c-TiO_2_(1Ag) samples had increased photoactivity compared to c-TiO_2_ samples. In contrast, p-TiO_2_(0.03Ag) and p-TiO_2_(1Ag) samples show similar or lower photodegradation rates than p-TiO_2_ samples. This shows that the silver modification by impregnation method caused a better improvement in the photoactivity of c-TiO_2_ coatings, compared to p-TiO_2_ coatings. This was also confirmed by the photodegradation tests in different dye solutions. This is mainly caused by the differences of the pore structure, which results in different amounts of Ag particles after deposition. XPS and RBS results show that c-TiO_2_ samples were capable of adsorbing more silver than p-TiO_2_ coatings (see Table 2). This is most likely because the c-TiO_2_ samples have a more open and impregnable pore structure, as it is also seen from the dye adsorption results (see Section 3.2.1).

It can also be assumed that there is an optimal amount of silver particles for a given type of TiO_2_ coating, and too high surface coverage can be disadvantageous by decreasing the accessible TiO_2_ surface. This was observed in the case of both p-TiO_2_ and c-TiO_2_ type samples, as the coatings impregnated in 0.03M AgNO_3_ solutions often showed better results than those impregnated in 1M AgNO_3_ solutions. TEM images with EDS elemental maps of Ag (see Section 3.1) show that p-TiO_2_(1Ag) samples contained a significantly higher amount of Ag particles compared to the p-TiO_2_(0.03Ag) samples.

The p-TiO_2_–Ag coatings made by a different silver modification method (mixing directly into the precursor sol) showed very different results from the other coatings: these samples showed the highest photoactivity under UV light, but the lowest under visible light, compared to the other samples. Their wetting properties are also slightly different from the other coatings, which behaved mainly similarly to each other: p-TiO_2_–Ag coatings showed slower response to UV light. Most likely these significant differences are partly due to the much different pore structure that could form in these coatings (see Section 3.1. TEM results): while the p-TiO_2_(1Ag) and p-TiO_2_(0.03Ag) samples show a structure made of smaller TiO_2_ particles forming a structure with at most 5–8 nm sized pores, the p-TiO_2_–Ag samples contain bigger TiO_2_ particles surrounding wider pores of 10–15 nm. The other main reason for their distinct behavior is the applied silver modification method, which lead to different silver content and distribution in these samples. In the p-TiO_2_–Ag coatings the total silver amount was much higher than in all the other samples according to XPS and RBS results (Table 2). However, according to TEM images and EDS Ag-maps (Figure 2) the surface is covered with very small (0.5–1 nm) sized particles. This suggests that the silver is mainly incorporated in the TiO_2_ crystal structure or dispersed on the surface as very small clusters and particles. The high photoactivity observed under UV light shows that a high amount of Ag content, with Ag dispersed finely on the surface and also present in the titania matrix, was advantageous in these photocatalysis tests. However under visible light the p-TiO_2_–Ag samples showed the lowest photoactivity, highlighting how silver modification can have different effects on the photoactivity depending on the mechanism of photodegradation and the measurement method. Initially, for these p-TiO_2_–Ag samples, the dimer peak increased, suggesting that monomer-to-dimer transformation occurred during the measurement, which might also contribute to the initially much slower degradation. This might be caused by the different pore structure of p-TiO_2_–Ag samples, as the dimerization can occur more easily in the bigger pores of these coatings.

Interestingly, for MB dye, the above described comparisons between the samples were no longer true. Silver modification by impregnation method did not improve the photoactivity, not even for the c-TiO_2_ coatings. The initial increase of the trimer peak shows that dye association happened in all coatings under UV irradiation, and this increase was more significant in the silver modified coatings, meaning that the presence of silver facilitated the association process. Trimers show much higher photostability, and thus this trimerization process significantly decreased the photodegradation rates of MB dye. By comparison, the photodegradation tests in MB dye solution showed the advantageous effect of the presence of silver, since in solution the dye association process is less significant than in the small pores of the coatings. This shows that there are processes (e.g., association of the organic molecule) that can significantly influence the efficiency of a self-cleaning photocatalyst surface, but cannot be studied by conventional dye solution photodegradation tests.

## 5. Conclusions

Photocatalysis tests at the liquid–solid interface showed that silver modification significantly increased the photoactivity of the TiO_2_ coatings. However, this was not always true in the case of air–solid interface measurements: for some samples with higher silver content, these tests even showed decreased photoactivity. This was especially significant for the tests using methylene blue dye, where dye association processes were more important. Most likely this is because if the surface coverage by Ag particles is too high, there is not enough titania surface available to the dye molecules, and thus the photodegradation rate decreases. Surface coverage by silver, presumably, is less important at the liquid–solid interface, since in that system the mobility of the dye molecules is much higher in the liquid filled pores.

Pore structure was found to influence the photoactivity indirectly: First, the amount of silver that is taken into the pores during the impregnation depends on the pore sizes (coatings with smaller pores were found to take up higher amount of silver, and thus the photoactivity increase was more significant). Second, the extent of dye association—which can hinder the photodegradation—depends strongly on the type of dye, but it is also significantly influenced by the pore structure of the coatings. Furthermore, the presence of silver in the pores also had an important influence on dye association, in two different ways. It was found that during dye accumulation the dye molecules would adsorb in monomer and associated forms (dimers, trimers), and if silver particles were present then the degree of association was decreased. Since the associated forms show slower photodegradation rates, the overall dye degradation rate at air–solid interface increases if there are less associated dye molecules. In contrast, the dye association process that could also occur under light irradiation during the photodegradation measurements was more prominent in the presence of silver, which hinders the photodegradation. A possible explanation is that the presence of Ag particles on the surface inhibits the dye-pore wall interactions, which slightly increases the mobility of the dye molecules, thus leading to increased tendency to associate during the irradiation.

## Figures and Tables

**Figure 1 nanomaterials-11-02240-f001:**
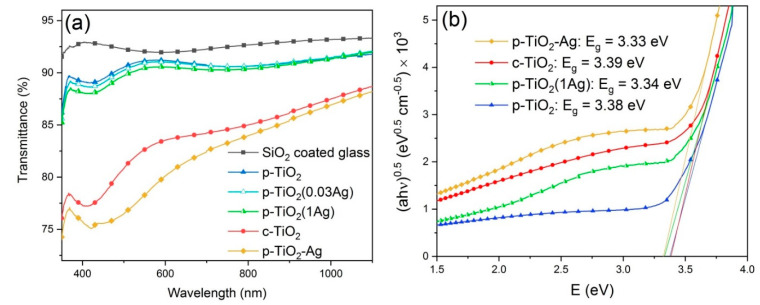
(**a**) Transmittance spectra of the SiO_2_ coated glass substrate, the different TiO_2_ coatings and the TiO_2_–Ag composite coatings on the example of p-TiO_2_ type samples; (**b**) Tauc plots of samples prepared on quartz glass.

**Figure 2 nanomaterials-11-02240-f002:**
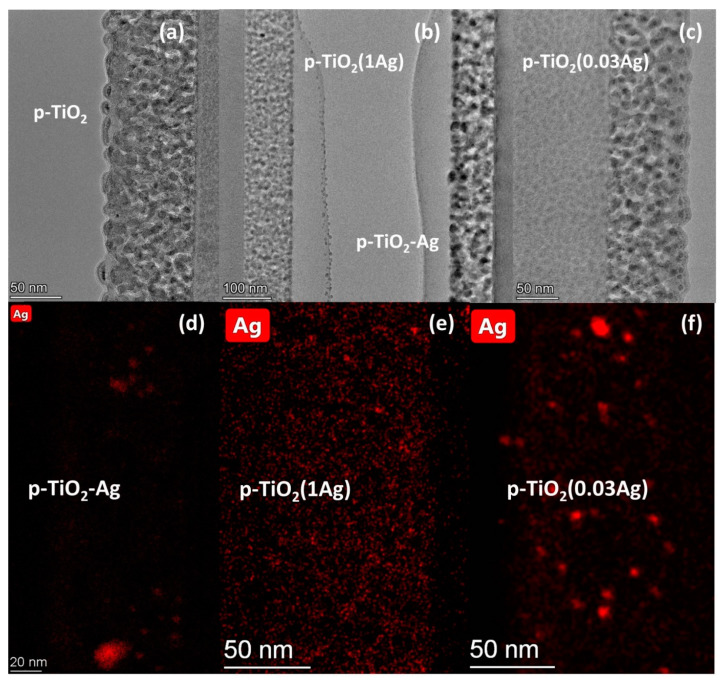
Cross-sectional HRTEM images of the (**a**) p-TiO_2_, (**b**) p-TiO_2_(1Ag) (**left**) and p-TiO_2_–Ag (**right**), and (**c**) p-TiO_2_(0.03Ag) samples. EDS elemental maps showing Ag particles of the (**d**) p-TiO_2_–Ag, (**e**) p-TiO_2_(1Ag) and (**f**) p-TiO_2_(1Ag) coatings.

**Figure 3 nanomaterials-11-02240-f003:**
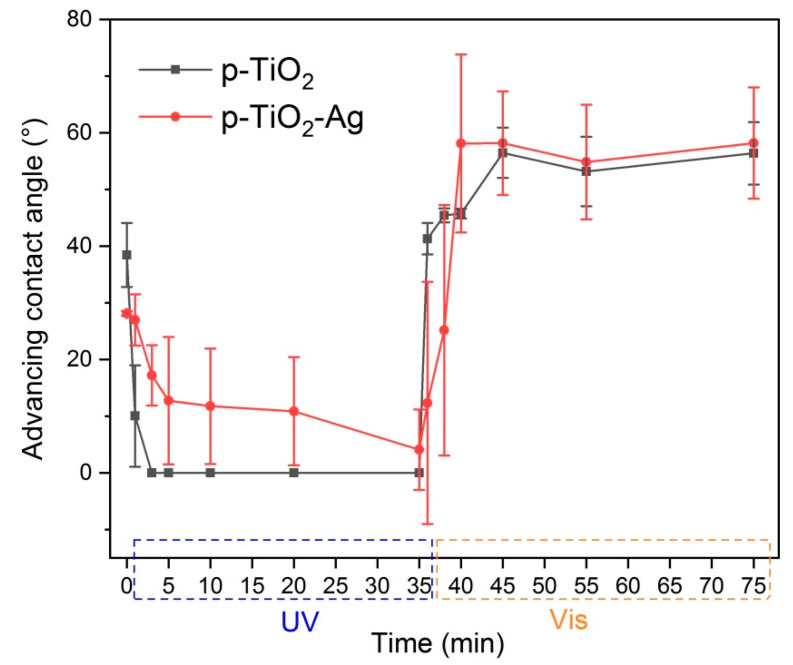
Photoinduced wettability conversion behavior of p-TiO_2_ and p-TiO_2_–Ag samples.

**Figure 4 nanomaterials-11-02240-f004:**
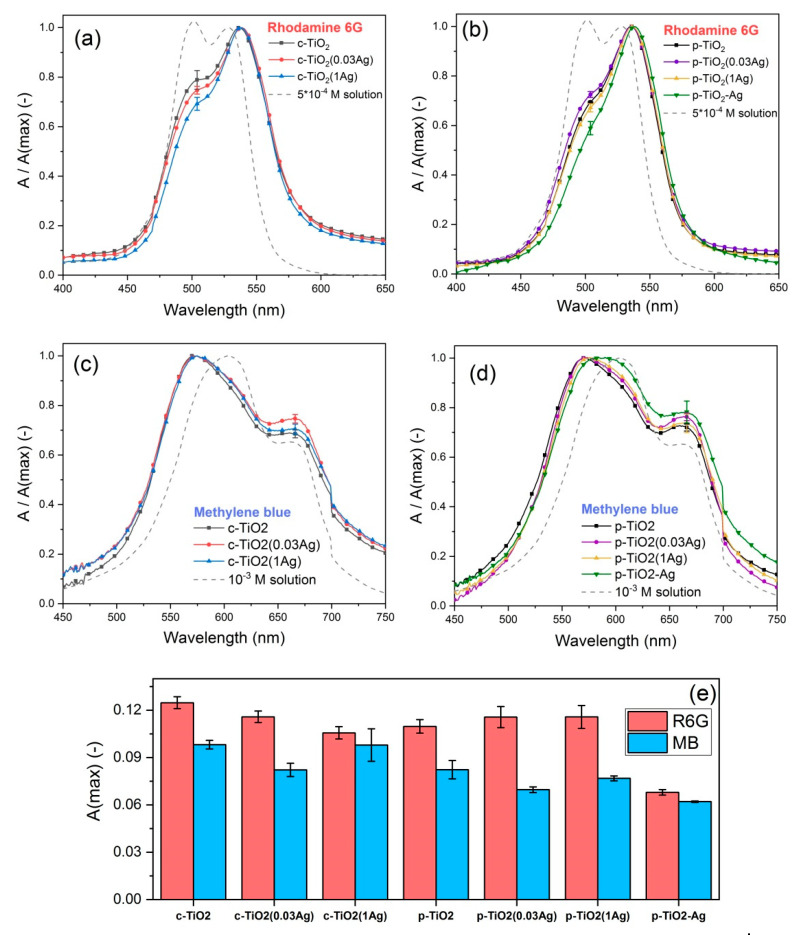
Normalized absorbance spectra of (**a**,**b**) Rhodamine 6G impregnated samples, and (**c**,**d**) methylene blue impregnated samples; (**e**) the comparison of maximum absorbance measured for the different dye impregnated samples.

**Figure 5 nanomaterials-11-02240-f005:**
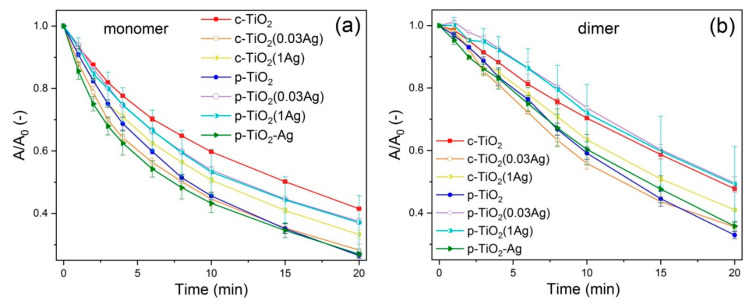
Rhodamine 6G dye degradation with UV illumination shown by *A/A_0_ − t* curves of (**a**) monomer and (**b**) dimer degradation. The monomer degradation is significantly faster than the dimer degradation.

**Figure 6 nanomaterials-11-02240-f006:**
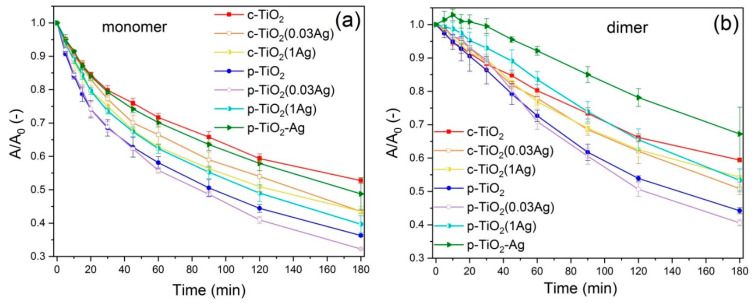
*A/A_0_ − t* curves of Rhodamine 6G (**a**) monomer and (**b**) dimer degradation under visible light.

**Figure 7 nanomaterials-11-02240-f007:**
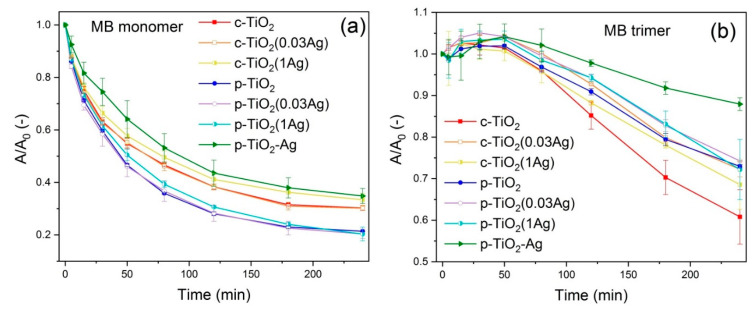
*A/A_0_ – t* curves of methylene blue (**a**) monomer and (**b**) trimer degradation under UV light. The monomer photostability is much lower compared to the associated form, while trimerization also occurs during the UV irradiation.

**Figure 8 nanomaterials-11-02240-f008:**
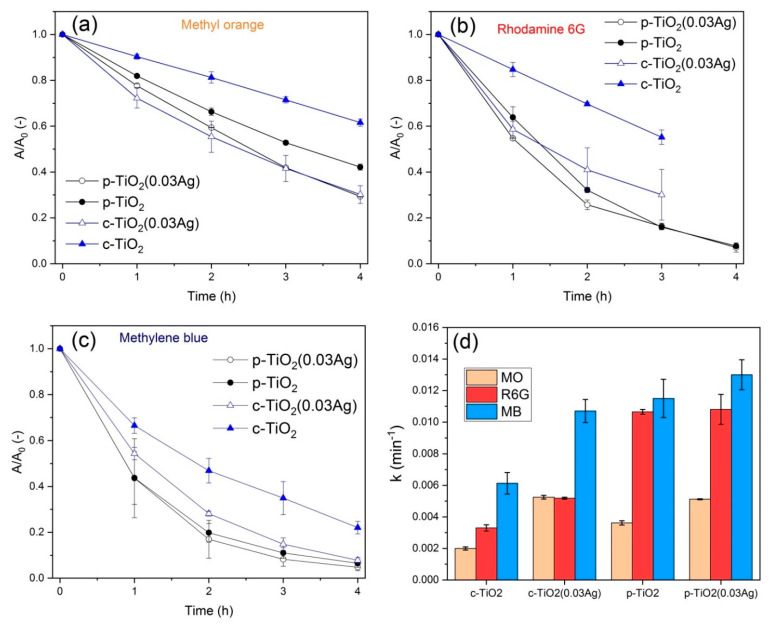
*A/A_0_ − t* curves of (**a**) methyl orange, (**b**) Rhodamine 6G and (**c**) methylene blue dye photodegradation at the liquid–solid interface, in the case of different TiO_2_ samples; (**d**) kinetics of dye degradation: first order *k* values.

**Table 1 nanomaterials-11-02240-t001:** Refractive index (n), film thickness (d) and porosity (P) values of SiO_2_ barrier layer on glass, and the different TiO_2_ layers deposited on SiO_2_ coated glass. Average pore size values (d_pore_) are included from our previous publication [18], determined by ellipsometric porosimetry on samples prepared by dip-coating on Si wafers.

	n	d (nm)	P (%)	d_pore_ (nm)
SiO_2_	1.454 ± 0.007	196 ± 10	-	-
c-TiO_2_	1.860 ± 0.052	54 ± 4	27 ± 3	5.0
p-TiO_2_	1.540 ± 0.040	88 ± 6	49 ± 3	9.0

**Table 2 nanomaterials-11-02240-t002:** Expected silver content of the samples and determined by RBS and XPS measurements. The RBS silver content values for c TiO_2_(0.03Ag) and c-TiO_2_(1Ag) samples are taken from our previous publication [31].

	Expected Ag wt %	RBS Ag wt %	XPS Ag wt % Surface	XPS Ag wt % Bulk
p-TiO_2_(0.03Ag)	0.08	0.67	0.7	0.6
p-TiO_2_(1Ag)	2.67	1.99	1.7	1.6
p-TiO_2_–Ag	11.80	6.91	7.4	7.5
c-TiO_2_(0.03Ag)	0.03	1.07	1.2	0.5
c-TiO_2_(1Ag)	1.04	2.37	4.3	4.3

## Data Availability

The data presented in this study are available on request from the corresponding author.

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
