# Peer review of "Effect of Silver Modification on the Photoactivity of Titania Coatings with Different Pore Structures"

_nanomaterials, 2021, doi:10.3390/nano11092240_

Round 1
Reviewer 1 Report
In this manuscript, authors report the preparation of TiO2-Ag hybrid nanomaterials and study the effect of silver modification on the photoactivity of titania coatings
Overall speaking, this work present good results, however some revision is needed
It is recommended that authors considered some articles on TiO2-Ag hybris materials see for example Applied Surface Science 2014, 288, 503-512 or some of the works of Zaleska (not included in Reference 7) group see for example Separation and Purification Technology 2010, 72, 309-318
Figure1: resolution of HRTEM is too low, new micrographs are need to characterize and observe the size of the nanoparticles
Table 2, need to have uniform significant figures and cut in the 2 digits after dot. In XPS is Ok with 1 digit. Also, in Table S3, included just relevant digits
As mentioned already, the manuscript indeed discusses good work however it needs some revision to be considered for publication in Nanomaterials.
Reviewer 2 Report
This manuscript describes synthesis of Ag-doped TiO2 sol-gel coatings. The research is of a high quality and well structured. The samples are characterized by modern methods. I have only a few minor remarks.
- XRD patterns of the samples should be provided in the main text.
- Fig. 2. The mapping is not clearly shown, the quality of the figure should be improved.
